# Local Response and Barrier Recovery in Elderly Skin Following the Application of High-Density Microarray Patches

**DOI:** 10.3390/vaccines10040583

**Published:** 2022-04-10

**Authors:** Fredrik Iredahl, David A. Muller, Totte Togö, Hanna Jonasson, Ben Baker, Chris D. Anderson, Joakim Henricson

**Affiliations:** 1Department of Primary Health Care, Region Östergötland, 58185 Linköping, Sweden; fredrik.iredahl@liu.se; 2Department of Medical and Health Sciences, Faculty of Health Sciences, Linköping University, 58183 Linköping, Sweden; 3School of Chemistry and Molecular Biosciences, The University of Queensland, Building 76 Cooper Road, St Lucia, QLD 4072, Australia; d.muller4@uq.edu.au; 4Department of Pain and Rehabilitation Center, County Council of Östergötland, 58185 Linköping, Sweden; totte.togo@regionostergotland.se; 5Department of Biomedical Engineering, Linköping University, 58183 Linköping, Sweden; hanna.jonasson@liu.se; 6Vaxxas Pty Ltd., Translational Research Institute, 37 Kent Street, Woolloongabba, QLD 4102, Australia; bbaker@vaxxas.com; 7Department of Biomedical and Clinical Sciences, Faculty of Health Sciences, Linköping University, 58183 Linköping, Sweden; chris.anderson@liu.se; 8Department of Emergency Medicine, Local Health Care Services in Central Östergötland, 58185 Linköping, Sweden

**Keywords:** microneedles, vaccination, skin barrier integrity, evaporimetry, skin reactivity, polarisation spectroscopy, dielectric permittivity

## Abstract

The high-density microneedle array patch (HD-MAP) is a promising alternative vaccine delivery system device with broad application in disease, including SARS-CoV-2. Skin reactivity to HD-MAP applications has been extensively studied in young individuals, but not in the >65 years population, a risk group often requiring higher dose vaccines to produce protective immune responses. The primary aims of the present study were to characterise local inflammatory responses and barrier recovery to HD-MAPs in elderly skin. In twelve volunteers aged 69–84 years, HD-MAPs were applied to the forearm and deltoid regions. Measurements of transepidermal water loss (TEWL), dielectric permittivity and erythema were performed before and after HD-MAP application at *t* = 10 min, 30 min, 48 h, and 7 days. At all sites, TEWL (barrier damage), dielectric permittivity (superficial water);, and erythema measurements rapidly increased after HD-MAP application. After 7 days, the mean measures had recovered toward pre-application values. The fact that the degree and chronology of skin reactivity and recovery after HD-MAP was similar in elderly skin to that previously reported in younger adults suggests that the reactivity basis for physical immune enhancement observed in younger adults will also be achievable in the older population.

## 1. Introduction

As the world enters the third year of the COVID-19-pandemic, the emergence of new SARS-CoV-2 variants such as Omicron are expected to cause the continued need for additional vaccine boosters to protect risk groups [1]. At the beginning of the COVID-19 pandemic, age was quickly identified as a prominent risk factor [2], making vaccination of the elderly a priority. Typically, most vaccines are administered via subcutaneous or intramuscular routes [3]. However, these vaccine delivery methods have disadvantages linked to the requirement of administration by health care professionals, the generation of contaminated sharp waste, and the risk of local side effects related to injection [4]. An alternative route of vaccination is by microneedles, which aims at the intradermal delivery of the vaccine. We have experience with a high-density microarray patch (HD-MAP) vaccine delivery platform. The HD-MAP applied to the skin provides practical advantages, including the absence of sharps, a reduced dependence or total removal from the cold chain, and the possibility for self-application through the use of a disposable applicator [5,6,7]. The HD-MAP delivers the vaccine antigen to the epidermal and dermal layers of the skin containing a high density of dendritic cells. As a function of this targeted vaccine delivery, our group and other researchers have demonstrated that enhanced immune responses with lower doses of vaccine are achievable as compared to traditional needle-based vaccination methods [8,9]. Previous studies have demonstrated the broad utility of the HD-MAP delivery platform, which has shown promising results against the dengue virus [10,11,12], influenza [13,14], West Nile virus, Chikungunya virus [15], Japanese encephalitis [16], and poliovirus [17,18]. Recently, pre-clinical studies using the HD-MAP have shown promising results producing broadly high titre neutralizing antibody responses against the SARS-CoV-2 ancestral virus and variants of concerns [19].

To date, previously reported clinical studies evaluating microneedle vaccinations have been performed in young adults in whom the skin is relatively reactive to external stimulus/provocation resulting in adaptive and innate immune reactivity [20,21,22]. Previous experience of vaccination in the elderly, for instance against influenza, has shown that it is necessary to apply an increased dose of the vaccine to maintain efficacy [23,24]. The kinetics of skin barrier recovery following penetration by HD-MAP is relevant for subject acceptability. Quantification of transepidermal water loss (TEWL) by evaporimetry is a classic method for assessing the barrier function in the skin. While there is always a background level of water vapour escaping the surface, once the HD-MAP punctures the skin, the integrity of the *stratum corneum* is lost and the rate of water loss from the viable epidermis increases markedly. Thus, measurements of TEWL can serve as an indicator of successful HD-MAP engagement as well as being *per se* an illustration of the induction of an innate reactivity and of an individual’s wound healing kinetics after the minor trauma caused by the HD-MAP application. In younger individuals, we have previously observed skin barrier recovery within 48 h observed by TEWL [20], and the resistance to topical histamine application after the same time period [25]. Non-invasive methodology allows documentation of skin barrier penetration/damage, induction of reactive inflammation, and the possibility to follow changes over time, which are the advantageous attributes of microneedle vaccine administration, without the discomfort and possible minor side events of the skin biopsy technique.

The aim of this study was therefore to study the reactive capability of elderly skin to the application of uncoated (not containing vaccine) HD-MAPs to the skin of the volar forearm and deltoid region, quantifying reactivity by measuring transepidermal waterloss as an indicator of both barrier penetration and reactivity as well as superficial skin hydration and erythema as indicators of inflammatory response.

## 2. Materials and Methods

### 2.1. Volunteers

Twelve volunteers, with a mean age ± standard deviation (range min to max) of 75 ± 5 (69 to 84) years and a body mass index of 27 ± 5 (21 to 37) kg/m^2^, were recruited and gave their written informed consent. The volunteers were asked to refrain from drinking coffee, tea, or alcohol and from exercise on the day of each measurement. The volunteers were seated comfortably in a semi-supine position during the measurements, which started after 20 min of acclimatization. Provocations and measurements were undertaken at the forearm and deltoid regions. Lighting conditions in the room were kept constant by closing window blinds and turning off ambient lights during measurements with the polarisation spectroscopy device. The room temperature was maintained at 21 ± 1 °C. The non-invasive nature of the assessment methods facilitated the study of individual phenotype, variability due to the presence or severity of disease as well as the possible influence of interactivity between therapeutic agents and vaccine effectiveness. The study was approved by the Regional Ethics Committee at Linköping University Hospital (Dnr. MB 2017-409-31, 18 October 2017).

### 2.2. Procedure

Participants received one HD-MAP applied to the volar forearm and one applied to the deltoid region on the non-dominant arm (Figure 1), as previously described by Muller et al. (2020) [20]. The application sites were free from hair, tattoos, scars, and visible veins. All areas were cleaned with an alcohol swab prior to application. Before application, the sites were compressed with a pre-calibrated skin conditioning ring (Vaxxas Pty Ltd., Brisbane, Australia) at a force of 30 N, to equalise the resistance of the skin between volunteers. A preloaded applicator device (Vaxxas Pty Ltd., Brisbane, Australia) was then docked into a skin conditioning ring. The applicator was triggered to apply the HD-MAP at a speed of 20 m/s. Once applied, HD-MAPs were left in place for 2 min. Immediately after removal, pain and local signs, erythema, oedema, and the presence of minimal breakthrough or minor “wet” bleeding were assessed clinically by trained staff, and skin physiological measurements were performed. Wet bleeding was defined as the presence, immediately after patch removal, of multiple pinpoint blood spotting on an applied tissue paper. Perceived pain on a scale from 0 (no pain) to 10 (worst imaginable pain) was reported by the volunteers at 10- and 30-min post-HD-MAP application. Measurements were recorded in the order of transepidermal water loss (evaporimetry), dielectric permittivity, erythema (polarisation spectroscopy), and by regular, high-resolution photography (dermoscopy). Dermatoscopic and dielectric permittivity images were subsequently assessed at the study’s time points for the grading of the occurrence and degree of petechia, “black dots” (thought to be oxidation products in puncture points), scaling (dry flaking skin), and change in the hydration of dielectric permittivity image brightness at the HD-MAP site [20,26]. Skin physiological measurements were performed in the same order at 10 min and 30 min after application and at return visits at 48 h and 7 days.

### 2.3. Equipment

#### 2.3.1. High-Density Microarray Patches (HD-MAP)

The HD-MAP is a 1 × 1 cm patch containing 3136 conically shaped, from 120 µm at the bottom to 25 µm at the top, solid microprojections with a length of 250 µm. It is constituted by a non-dissolvable liquid crystal polymer and has a max penetration depth of 150 µm but an average penetration of around 100 µm. The HD-MAPs used were non-coated and applied as described in the procedure section.

#### 2.3.2. Evaporimetry

Transepidermal water loss was used to indicate penetration of the epidermal barrier. The measurement was performed by a Tewameter-300 probe (Courage + Khazaka electronic GmbH, Köln, Germany) at each application site. Evaporimetry measures transepidermal water loss as an indirect measure of barrier integrity. An increase in TEWL is expected when the barrier is compromised. At each measurement, the Tewameter probe was placed over the HD-MAP application site, and measurements were collected every second for 30 s and stabilised values selected for analysis. Collected data were analysed by CK-MPA-Multi-Probe-AdapterFB, v2.4.2.1/207-11-10 (Courage + Khazaka electronic GmbH, Köln, Germany).

#### 2.3.3. Dielectric Permittivity

An Epsilon™ device (Epsilon Modell E100, Biox, London, UK) was used to estimate the presence of water in the superficial layer, *stratum corneum*, of the skin. The measurement principle has previously been well described [27]. Briefly, the system utilises the measurement of the dielectric constant of the upper 5 µm of the skin to create a two-dimensional hydration image of the skin surface. Due to 76,800 sensors within the sensing area of 12.8 × 15 mm, skin surface hydration can be mapped. We used the standardised “burst mode” option with a 5 s delay after the first skin contact, a frame interval of 1 s, and a total measurement period of 30 s.

#### 2.3.4. Polarisation Spectroscopy

Tissue Viability Imaging (TiVi, WheelsBridge AB, Linköping, Sweden) was used to estimate erythema (*rubor*). The measurement principle has previously been described in detail [28]. Briefly, based on the wavelength-dependent absorption properties of red blood cells (RBCs) and imaging-processing algorithms, a TiVi-value (in arbitrary units, AU) that is linearly proportional to the local RBC concentration in the skin is calculated for each pixel. Image analysis and the calculation of TiVi values were made using WheelsBridge AB Software, v1.2.20, November 2018, Linköping, Sweden. The TiVi system was set to single photo mode and a medium resolution and positioned approximately 25 cm above the observed site.

#### 2.3.5. Dermoscopy

Photographic documentation of the application sites was performed using a dermatoscope (iC1, Heine Optotechnik GmbH & Co., Gilching, Germany) attached to a mobile telephone (iPhone 6, Apple, Cupertino, CA, USA). Dermoscopy refers to the examination of the skin using skin surface microscopy. Dermoscopy is mainly used to evaluate pigmented skin lesions but is also used for detailed analysis and documentation of inflammatory events.

#### 2.3.6. Statistical Analysis

Data are presented in terms of mean ± SD. One way ANOVA with Šidák’s multiple comparison test was used to compare changes between pre-application values and values at 10 min, 30 min, 48 h, and 7 days. Differences between the forearm and deltoid regions were tested with the Student’s *t*-test. The alpha level for statistical significance was set at 0.05. All statistical analyses were made with the aid of GraphPad Prism v9.1.2 for Windows (GraphPad Software, San Diego, CA, USA, www.graphpad.com (accessed on 1 March 2022)).

## 3. Results

HD-MAP applications caused minimal discomfort. Perceived pain on a scale from 0 (no pain) to 10 (worst imaginable pain) was reported by the volunteers at 10- and 30-min post-HD-MAP application. At 10 min, no volunteer reported perceived pain values higher than 3. At 30 min, all volunteers, except one that reported 2, reported 0 perceived pain (Appendix A). Minor wet bleeding was seen in 12 of the 24 HD-MAP applications immediately upon removal of the HD-MAP (Table 1). Petechiae were seen in most images at 10 and 30 min to be virtually absent at later time points. Black dots at skin puncture points had developed in about half the cases by 48 h and 7 days. The dermatoscopic images showed uneven dry and flaking skin in the pre-application images in 9 of 24 cases and almost twice as many at later time points (Table 1). A representative series of dermatoscopic images can be seen in Figure 2.

### 3.1. Quantification of Transepidermal Waterloss

Skin barrier integrity was indirectly assessed as TEWL (g/hm^2^) as measured by evaporimetry. Mean pre-application TEWL values did not differ significantly between regions (forearm; 6.65 ± 1.08 g/hm^2^, deltoid; 6.63 ± 2.19 g/hm^2^, *p* = 0.42). At the 10- and 30-min observations, TEWL was significantly increased compared to respective pre-application values (*p* < 0.0001) at both the forearm and deltoid (Figure 3A,B). The highest mean TEWL values were observed at 10 min, 67.50 ± 13.49 g/hm^2^ at the forearm and 66.01 ± 10.95 g/hm^2^ at the deltoid. At 7 days there was no significant difference compared to pre-application values at either region (forearm; *p* = 0.99, deltoid; *p* > 0.99).

### 3.2. Quantification of Superficial Skin Hydration

Superficial skin hydration, i.e., the presence of water in the *stratum corneum,* was assessed as dielectric permittivity. Mean pre-application values for dielectric permittivity did not differ significantly between the application sites (forearm; 5.86 ± 2.05 F/m, deltoid; 5.31 ± 2.42 F/m; *p* = 0.11). The dielectric permittivity increased significantly (increased *stratum corneum* hydration) during the first 30 min at both sites (24 of 24 HD-MAPs) compared to respective pre-application values (forearm; pre-application vs. 10 min, *p* < 0.0001, pre-application vs. 30 min, *p* < 0.0001; deltoid; pre-application vs. 10 min *p* < 0.0001, pre-application vs. 30 min *p* < 0.0001) (Figure 3C,D). The highest values for dielectric permittivity were observed at 10 min, 34.00 ± 12.23 F/m at the forearm, and 39.08 ± 18.53 F/m at the deltoid. At 48 h there were no significant differences between the pre-application values (forearm; pre-application vs. 48 h *p* > 0.99; deltoid; pre-application vs. 48 h, *p* = 0.99). There was no significant difference compared to the respective pre-application values at 7 days for either site (forearm; *p* = 0.76, deltoid; *p* = 0.88). However, decreased signals (decreased hydration) compared to pre-application values at both sites were observed in slightly less than half of the images at both 48 h and at 7 days (forearm; pre-application mean dielectric permittivity value 5.87 ± 2.054 F/m, 48 h; 6.15 ± 0.56 F/m; 7 days, 2.16 ± 0.56 F/m; deltoid; pre-application 5.31 ± 2.42 F/m; 48 h, 5.70 ± 2.3 F/m, 7 days 1.86 ± 0.45 F/m) (Figure 4).

### 3.3. Quantification of Red Blood Cell Concentration

The kinetics of the erythema reaction to HD-MAP application was quantified by TiVi. The mean TiVi value was 128.75 ± 43.51 AU at the forearm site and 126.08 ± 35.32 AU at the deltoid site and did not vary significantly between (*p* = 0.84). TiVi values at 10 min and 48 h were significantly increased (*p* ≤ 0.0001) at the forearm and deltoid sites compared to the respective pre-application values (Figure 3E,F). TiVi values returned to pre-application levels for both test sites at 7 days (forearm; *p* = 0.45, deltoid *p* = 0.99). TiVi values peaked at 30 min (275.1 ± 66.45 AU at the forearm site and 240.2 ± 63.19 AU at the deltoid site) with slightly lower values at 10 min (forearm; 260.6 ± 57.96 AU, deltoid; 233.3 ± 56.58 AU).

## 4. Discussion

Here, we reported the innate reactivity and recovery of the skin in older adults following the application of HD-MAPs. The kinetics of the reaction were similar to those previously reported in younger skin [20]. Using a range of complementary skin physiological methods of investigation, we found that HD-MAP application was associated with an instant loss of barrier integrity as measured by transepidermal water loss, which implies capacity for vaccine delivery. HD-MAP application also triggered a rapid increase in erythema and surface hydration, which indicates the inflammatory provocation of the skin as a result of direct cell damage or the axon reflex mechanism [20,29]. The application of the HD-MAP resulted in erythema at 48 h, indicating tissue reactivity at a later time point than the rapid, direct cellular damage or axon-reflex mediated reaction. Evaporimetry and surface hydration values began by 48 h to approach pre-application values with near complete recovery within a week, except for surface hydration which fell to lower levels than at the outset. Decreased hydration fits with the dermoscopy images showing flaking skin, and we interpreted this as post inflammatory desquamation, a phenomenon not observed in younger skin. With the exception of the flaking skin, the results were similar chronologically and in absolute values to previously observed data in the younger individuals. The present findings support the capacity for HD-MAP application to cause inflammatory reactivity in the skin, thought to be one of the primary drivers of the enhanced immune response observed following microneedle vaccination, even in elderly individuals. These findings suggest that the immune-enhancing phenomena associated with HD-MAP vaccine delivery will likely be seen also in the older age group.

The elderly skin did show some differences to the results seen in younger skin. The minor and transitory pinpoint wet bleedings observed after HD-MAP application in 50% of the present study’s volunteers was more than the 28% observed in a previously published younger population [20] after HD-MAP application. Likewise, petechiae were more common at early time points in this older group. The appearance at later time points of flaking, dry skin caused, we believe, some subtle changes in the development of the black dots, which indicate points of microneedle penetration. This is likely due to structural changes in older skin [30,31] which may also result in mechanical factors of relevance to the HD-MAP application. The measurement of the inflammatory component erythema (*rubor*) by reflectance spectroscopy showed a similar reactivity to that previously observed in younger skin. In fact, the reactive erythema observed by reflectance spectroscopy in our aged group distributed itself above the mean values for the younger group previously studied. The fact that the occurrence of petechiae was more frequent in the older group may have contributed to the higher TiVi signal since the method quantitates total blood concentration in the tissue. In addition to the structural changes of elderly skin, a low-grade increase in the basal inflammatory state in aging skin, “inflammaging” is a hypothesised characteristic of elderly skin, which could be a functional etiological explanation [32,33,34]. Other groups have studied different aspects of the effects of microneedle application, often with devices of differing attributes to our HD-MAP. Points of focus have been micropuncture attributes and chronology of closure. Other delivery platforms such as microneedle patches [22] or dissolving microneedle patches [21] have likewise shown promising results regarding safety and tolerability. In our study, evaporimetry, dielectric permittivity, and polarisation spectroscopy were applied to observe different aspects of HD-MAP application. We chose evaporimetry because of its ability to observe epidermal penetration resulting in water loss. An alternative would have been OCT [35,36]. In our future work, we plan to perform direct comparative studies of young and old skin to better understand this matter further.

This study was limited by the fact that indirect physiological signs of inflammation were measured instead of direct molecular markers. Non-invasive methodologies facilitate, however, the performance of studies on both healthy individuals and individuals with disease or on specific medication, without the need for invasive methods for the direct measurement of biomarkers. The possibility exists for the combination of skin physiological techniques with minimally invasive techniques such as biopsy or other sampling methods with a view to establishing a robust relationship to more convenient and broadly applicable surrogate skin physiological methods for use in method development and studies of efficacy. Another limitation is that only 4 of 12 subjects returned on the 7-day revisit. The fact that our previous study had shown that tissue recovery was obvious by 7 days makes it less likely that the smaller size of the number of observations we had data on at 7 days would have hidden a lack of recovery.

## 5. Conclusions

Since the reactivity to HD-MAP application in this group of elderly volunteers (mean age 75 years) was similar to that seen in a previously published younger group (mean age 27 years), we conclude that skin reactivity in the older age group was maintained at a level which is likely sufficient to induce the physical adjuvance hypothesised to be one of the positive attributes of HD-MAP vaccination. A further detailed study of how skin reactivity is influenced by device features and application method, as well as the variability associated with subject phenotype, disease, or medication, is warranted.

## Figures and Tables

**Figure 1 vaccines-10-00583-f001:**
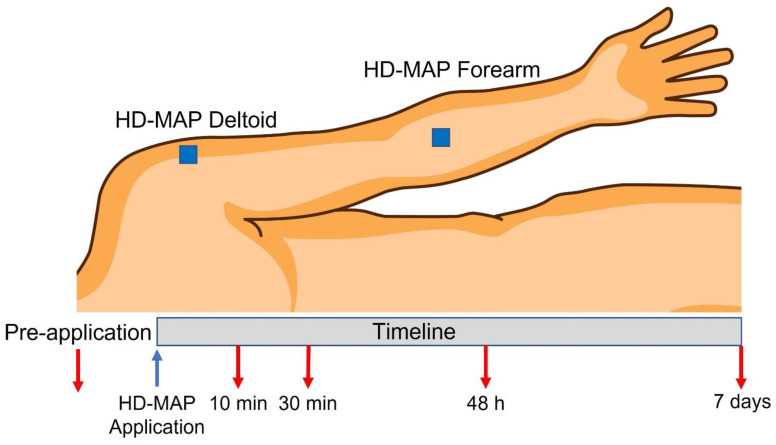
Schedule of the application sites of high-density microneedle array patches (HD-MAP). One HD-MAP applied to the volar forearm and one HD-MAP applied to the lateral deltoid region. Underline showing timepoints for measurements after the application of HD-MAP.

**Figure 2 vaccines-10-00583-f002:**
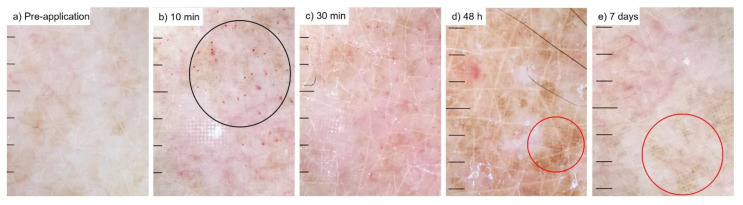
Typical dermatoscopic images from each time point ((**a**) pre-application, (**b**) 10 min after application of HD-MAP, (**c**) after 30 min, (**d**) 48 h and (**e**) 7 days). Note that petechiae are prominent after 10 and 30 min, and black dots are observed after 48 h and 7 days. The black circle displays areas with petechiae. Red circles display areas with black dots.

**Figure 3 vaccines-10-00583-f003:**
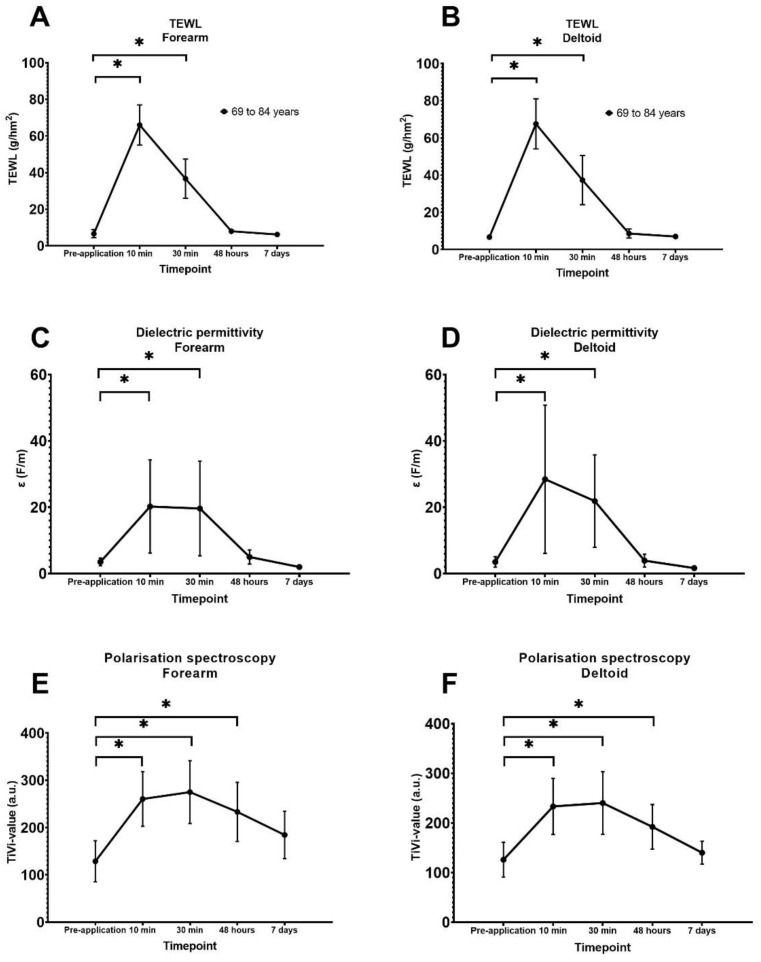
Graphs of mean values and standard deviation after measurement by evaporimetry (transepidermal water loss (TEWL)), dielectric permittivity, and polarisation spectroscopy (TiVi values) during the timepoints; pre-application, 10 min (*n* = 12), 30 min (*n* = 12), 48 h (*n* = 12), and 7 days (*n* = 4) after HD-MAP application. Explanations for subfigures: (**A**)—Forearm TEWL, (**B**)—Deltoid TEWL, (**C**)—Forearm polarisation spectroscopy, (**D**)—Deltoid polarisation spectroscopy, (**E**)—Forearm dielectric permittivity, (**F**)—Deltoid dielectric permittivity. * Shows significant difference with pre-application values.

**Figure 4 vaccines-10-00583-f004:**
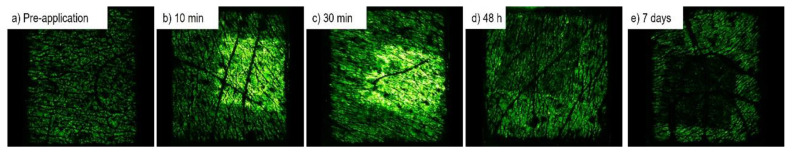
Typical dielectric permittivity images of one individual from each time point ((**a**) pre-application, (**b**) 10 min after application of HD-MAP, (**c**) after 30 min, (**d**) 48 h, and (**e**) 7 days). Note that an initial increased response is observed within 30 min after application, followed by a decreased response compared to pre-application after 48 h and 7 days.

**Table 1 vaccines-10-00583-t001:** The number of HD-MAP application sites with petechia, black dots, flaking, and minor wet-bleeding at different observations. Each individual received two HD-MAPs. Only data from 4 of the 12 volunteers were available at the 7 days follow up. Reactions grades as 1 = mild, 2 = moderate, 3 = severe, = not observed.

	Petechiae				Black Dots				Flaking				Wet Bleeding	
Grade	1	2	3	0	1	2	3	0	1	2	3	0	1	0
**Pre-treatment**	-	-	-	24	-	-	-	24	7	2	-	15	-	24
**10 min**	13	9	-	2	-	-	-	24	13	1	-	10	12	12
**30 min**	12	8	-	4	1	-	-	23	13	2	-	9	-	24
**48 h**	1	1	-	22	10	3	-	11	14	2	-	8	-	24
**7 days**	-	-	-	8	7	-	-	1	7	-	-	1	-	8

## Data Availability

The data presented in this study are available on request from the corresponding author. The data are not directly accessible in order to ensure correct interpretation of raw data characteristics.

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
