# Peer review of "Local Response and Barrier Recovery in Elderly Skin Following the Application of High-Density Microarray Patches"

_vaccines, 2022, doi:10.3390/vaccines10040583_

Round 1
Reviewer 1 Report
The focuses of the manuscript were on the local response and barrier recovery after applying high-density microarray patches (HD-MP) onto human skin. The participants are adults aged greater than 65 year old. In the previous studies, they were adults aged between 18-64. Similar results were found in both studies. However, more tests should be done as previous studies have demonstrated that mechanical stress, induce inflammatory response and cell death were observed.
Reviewer 2 Report
Although the paper is well presented and well writen, I have some concerns with the evaluation of the presence of inflammatory markers; the author do confess in their discussion that this later was not addressed in the current study. I woud suggest to address this point and compare them with inflammatory responses of the young skin.
Reviewer 3 Report
In this paper Iredahl et al. have reported local response and barrier recovery in elderly skin after application of high density microneedle patches.
The study is well designed and adequate experiments are condicuted to support the conclusions. The findings are of interest and would add to our understanding of effect of microneedle usage in elderly subjects
I have no specific comments and recommend acceptance
Round 2
Reviewer 1 Report
The revised manuscript is similar to the first version. A revised vresion should have descriptions on the following issues:
1) the comparison between the invasive and non-invasive methods, on the physiological change of skin (in the Introduction section),
2) the reason for choosing the non-invasive method (in the method section),
3) the studies on the young skin should be stated as the future work.
Reviewer 2 Report
Detail points raised in my previous review have been adressed adequatly.
